# Analysis of the Content and Comprehensiveness of Dermatology Residency Training Websites in Taiwan

**DOI:** 10.3390/healthcare9060773

**Published:** 2021-06-21

**Authors:** Po-Yu Chen, Ying-Xiu Dai, Ya-Chuan Hsu, Tzeng-Ji Chen

**Affiliations:** 1Department of Family Medicine, Taipei Veterans General Hospital, Taipei 112, Taiwan; barry50710@gmail.com; 2School of Medicine, National Yang Ming Chiao Tung University, Taipei 112, Taiwan; daiinxiu@gmail.com; 3Department of Dermatology, Taipei Veterans General Hospital, Taipei 112, Taiwan; 4Department of Family Medicine, Kinmen Hospital, Ministry of Health and Welfare, Kinmen 891, Taiwan; ych97160@gmail.com

**Keywords:** dermatology, residency website, residency training

## Abstract

With a growing trend in the popularity of web-based resources, it is important to evaluate residency program websites for providing accurate information for dermatology residency applicants. Little is known about the quality of dermatology residency websites in Taiwan. The aim of the study is to assesses the quality of official websites of dermatology training programs in Taiwan. A literature search for all related studies from inception to 31 July 2020 was performed using PubMed without restriction on language. We used criteria that had 6 domains and 25 items to evaluate 23 official websites of the dermatology training programs in Taiwan from August to September 2020. Of the 23 training programs, only 6 (26%) of the websites met more than half of the criteria. Notably, the items “features of the department” and “comprehensive faculty listing” were included in all websites. The criteria for interview process, board pass rates, social activities and information on the surrounding area were not met by all websites. Evidently, there is much room for improvement for the dermatology training program websites in Taiwan.

## 1. Introduction

### 1.1. Health Information on the Internet

In modern society, the Internet has gradually become more and more important for humans. According to statistics from the International Telecommunication Union, the percentage of Internet users worldwide increased from 16% in 2005 to 53.6% in 2019 [1]. In Taiwan in 2019, 92.78% of people used the Internet. It has become a convenient and generalized tool for acquiring information in Taiwan. A previous study reported that 80% of Internet users use the Internet to acquire healthcare information [2], and medical staff search it for information they need. Online resources for medical knowledge have become available to medical students and patients through computers and smart phones [3]. For physicians, clinicians and students, online resources represent an extensive source of continuing education in different specializations [4,5,6]. Patients can find several educational materials on the Internet, especially professional websites. In fact, a previous study showed that online patient education materials from the websites of major dermatology associations, such as the American Academy of Dermatology, was written well and appropriate for the general population [7]. Indeed, Internet has become a rather significant tool for physicians, medical students and patients.

### 1.2. Taiwanese Residency Training Programs: Dermatology

Since 2019, in Taiwan, medical students have been required to obtain a medical license after graduation and to finish postgraduate year training (PGY) within 2 years if they decide to receive advanced residency training [8]. Thereafter, they need to choose a specialization and complete residency training; for this purpose, they apply to several programs all over Taiwan, but many young doctors in the PGY-2 level are unfamiliar with the processes. Of all 23 specializations, dermatology is one of the most popular in Taiwan. The training period for dermatology residency is 4 years. Due to the advance of aesthetic medicine and the national health insurance system, there is tough competition to be accepted into a dermatology training program. Approximately 1400 medical students are able to obtain their medical license every year, but only about 28 positions per year are available for dermatology residency training. Even though this capacity can be adjusted by an agreement between the Taiwanese Dermatological Association and the Ministry of Health and Welfare, it is still much less than other specializations [9].

In the past, information on the training programs was difficult to obtain. Applicants often have to ask their senior residents or teachers for information or search the public information on the program websites. Applicants are less competitive without knowing people in dermatology departments. With widespread use, the Internet has become a common source of information for medical students applying for residency. Previous studies have indicated that the website content influenced an applicant’s decision-making, especially for those who had never been engaged in the department [10]. Prospective applicants rely heavily on the websites of residency training programs to obtain significant information regarding the application process, as well as the unique aspects of the different programs [11]. Therefore, the importance of maintaining an informative and accessible website continues to grow. Many studies abroad have evaluated the website content of different specialties, including dermatology [12,13,14,15,16,17,18]. However, there had been no research that studied the contents of websites on residency training programs in Taiwan. In this study, standardized criteria were used to evaluate the websites of dermatology training programs in Taiwan. It is expected to provide dermatology programs information about how to enrich the content and display the strengths of their program to applicants. By enhancing their websites, the training programs also can attract more excellent applicants.

## 2. Materials and Methods

### 2.1. Establishing the Criteria

We searched for previous research that evaluated the websites of different residency or fellowship programs through the online literature database PubMed, using the keywords “Internet standards,” “internship and residency,” “fellowship and scholarship,” and “information dissemination” from April to May 2020. We found 6 studies that performed questionnaire surveys to understand what content is important for applicants. After reevaluation, we developed a set of evaluation criteria with 25 content items categorized into 6 domains: recruitment information, department information, education and research, clinical work, incentives and frequently asked questions and answers (Table 1) [10,11,19,20,21,22,23]. The process of establishing the criteria is shown in Figure 1. The evaluation criteria were published previously in the *Taiwan Journal of Family Medicine* [23]. This is the first study to use these criteria to evaluate residency program websites in Taiwan.

### 2.2. Internet Search for Taiwanese Dermatology Training Program Websites

Thereafter, using the keywords “dermatology” and “training capacity,” we performed Google search for the online official website of the Taiwanese Dermatological Association to check for the list of qualified dermatology training hospitals. We linked our search to the website of the Taiwan Ministry of Health and Welfare to review the publication list of qualified dermatology training hospitals and the training capacity after revision in July 2020 [9]. The number of items that met the criteria was calculated for each website. We conducted the evaluations twice with different reviewers. The first review was conducted by the first author around August 2020. A second review was conducted by the second author around September 2020, blinded to the results of the first review. Since this study did not involve human subjects, it did not need institutional review board approval.

## 3. Results

### Website Evaluation

Overall, 23 dermatology residency training programs were approved by the Taiwan Ministry of Health and Welfare, and their official websites were evaluated in this study. The name of the training programs/departments and their website links are listed in Appendix A. Based on the common administrative regional classification in Taiwan, the distribution of the training programs among the 23 hospitals was as follows: 56.6% (13 programs) in the northern area, 17.4% (4 programs) in the central area, 21.7% (5 programs) in the southern area and 4.3% (1 program) in the eastern area. Of the 23 hospitals, 39.1% (9 programs) were public and 60.9% (14 programs) were private.

In comparison with the results of 2 surveys, there was no difference. The final results are listed in Table 2. The websites met a mean and median of 13 of 25 items (52%) of the criteria, with a maximum of 18 items (72%) and a minimum of 4 items (16%). Over half of the criteria were met by 26% (6 of 23 programs) of the dermatology departments, of which 3 were at public hospitals and 3 were at private hospitals. However, the dermatology websites of Cathay General Hospital, Taipei City Hospital, Renai Branch and Heping Fuyou Branch met only four items (16%) of the criteria.

The distribution of the items on the websites is listed in Table 3 and Figure 2. Most of the websites (over 85%, 20 of 23 programs) contained information on the following 4 items: features of the department (100%, 23 programs); comprehensive faculty listing (100%, 23 programs); description of the environment and equipment (91.3%, 21 programs); and history of the department (87%, 20 programs). Notably, the items on interview process, board pass rates, social activities and information on the surrounding area were not posted on any dermatology website.

We further analyzed and compared the proportion of public and private programs/departments that contained each item in our criteria. Public and private programs had the tendency to show similar information on most items (*p* > 0.05) in the 6 domains. These results are listed in detail in Table 3.

## 4. Discussion

### 4.1. Analysis of Website Contents

Previous related studies, most of which were from the United States of America (USA) and Canada, have been conducted to evaluate the websites of residency and fellowship specializations [14,15,16,17,18,19,20]. To our best knowledge, this study was the first to evaluate the content of dermatology residency program websites in Taiwan. Our study found highly variable content and quality among these websites. Most programs need improvements in the functioning of their webpages.

Among the criteria items, the most common information addressed were “features of the department” and “comprehensive faculty listing.” Through this, people can learn more about the development of the department and choose which doctor they prefer. Applicants can also learn of senior role models for their medical careers. Over half of the websites included “list of current residents/fellows,” but only 21.7% mentioned “alumni information and outcomes.” This information could show close relationships between the department and its staff, but it was not valued by most programs. The items of “program goals,” “research activities and accomplishments” and “educational resources available to residents” were included in more than half of websites. Information on the research achievements and education resources allow applicants to compare different activities and resources. Patients and the general public can also improve their medical knowledge.

Among the least common and poorly represented information were program-related specifics, such as “recruitment criteria,” “message from the chairman/program director,” “salary,” “interview process,” “board pass rates,” “social activities,” and “information on the surrounding area.” In most websites, announcements on the “recruitment criteria” and “interview process” appeared only during the period of application; at other times, we could not find the relevant information. Simple statements on recruitment would encourage more excellent applicants to apply and provide them with an understanding of the process. Notably, “message from the chairman/program director” was often neglected among the websites evaluated in this study but had been more frequent in foreign websites [13,18].

Salary is an item that most people and organizations do not disclose to the public in Taiwan. Most applicants might presume that the directors would think that these issues are not directly related with clinical training: “if you ask about these, it means you are only interested in the money and benefits, and do not have a passion for the medical training.” This observation had been reported in the USA and Taiwan [16].

Information about clinical work for residents was less clear on the internet. Information on “work hours,” “expected caseload,” and “on-call expectation” was provided in fewer than 10% of websites. Recently, the case loading and work hours of residency have been hot topics due to cases of death from overwork. Since residency doctors were included in the regulations of the 2019 Taiwanese Labor Standards Act, programs have been required to follow the Guideline of Residents Labor Rights Protection and Working Hours. Disclosure of this information is important for applicants [24].

### 4.2. Comparison with Research from the USA

Prior research includes one 2016 study about dermatology residency program websites in the USA [15]. We compared this with our study to understand the differences and found the following similarities and differences. Similarities with the USA study: program contact information, 83.5%; program description, 95.7%; research opportunities, 86.1%; current resident listing and 64.3%; current faculty listing, 89.6%. Our results were also all >50%. Other percentages >50% in the USA study were rotations and electives, 87.8%; applicant information, 88.7%; interview process, 64.3% (all >50%). In our study, these items were <40%. Most dermatology residency websites in the USA and Taiwan included content that described programs and listed faculty, residents and research accomplishments. However, Taiwanese websites rarely had information about recruitment, interviews and rotation. The disparity suggested much room for improvement in the website content in Taiwan.

### 4.3. Limitations

There were several limitations in our study. First, the criteria were according to foreign studies and may not be applicable to programs in Taiwan. In addition, the reliability and validity need to be verified further in the future. Second, our results represented a single snapshot in time. Owing to differences in navigability among the websites based on individual subjectivity, some information may have been overlooked during data collection. However, the likelihood that the differences in collection would change the nature of the presented data was low, even is this study was primarily descriptive in nature. Finally, our study focused only on the content of the websites and did not assess website design and quality. Each website has a unique design; therefore, quality can be difficult to compare. This aspect may be further analyzed in future.

## 5. Conclusions

Most of the dermatology websites from Taiwan did not contain sufficient information, especially recruitment, board pass rate, alumni, social activities, surrounding area, clinical work, salary, ancillary benefits and frequently asked questions and answers were especially neglected.

## Figures and Tables

**Figure 1 healthcare-09-00773-f001:**
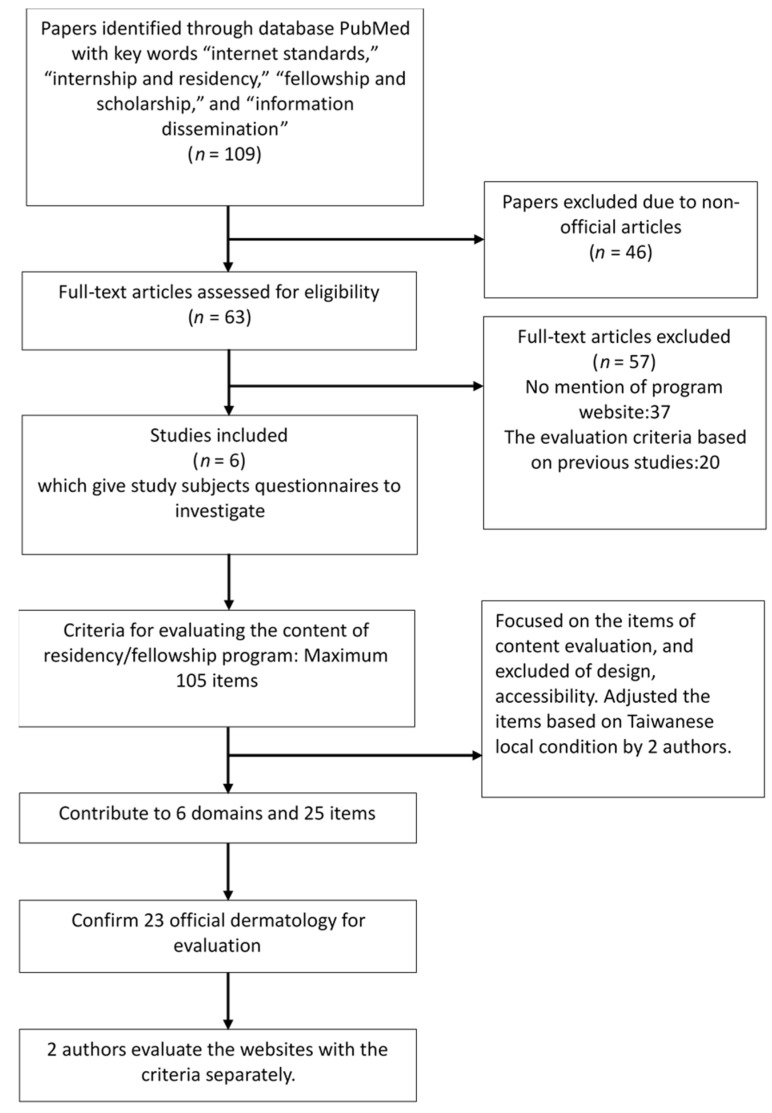
Flowchart of establishing the criteria and evaluating the websites.

**Figure 2 healthcare-09-00773-f002:**
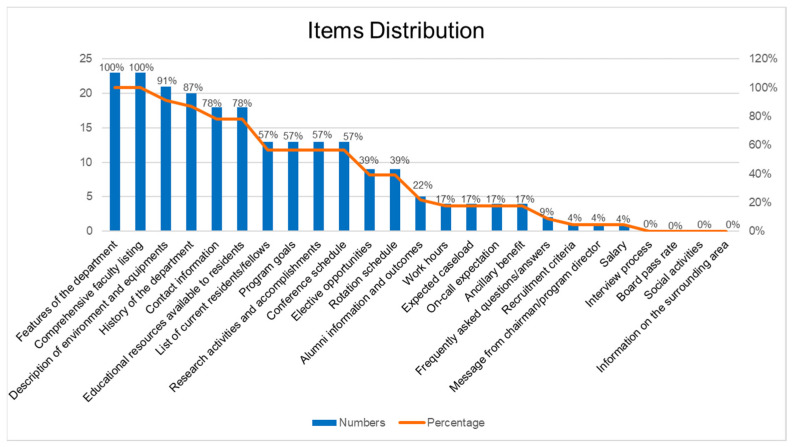
Item distribution over all programs.

**Table 1 healthcare-09-00773-t001:** Criteria for evaluating the websites of residency and fellowship programs.

Domains (6)	Items (25)
Recruitment information	Recruitment criteria
	Interview process *
	Contact information
Department information	History of the department
	Features of the department
	Comprehensive faculty listing
	Message from the chairman/program director
	List of current residents/fellows
	Board pass rates
	Alumni information and outcomes
	Description of environment and equipment
	Social activities
	Information on the surrounding area ^†^
Education and research	Program goals
	Research activities and accomplishments
	Educational resources available to residents
	Conference schedule
	Elective opportunities
Clinical work	Work hours
	Rotation schedule
	Expected case load ^§^
	On-call expectation
Incentive	Salary
	Ancillary benefit ^||^
Frequently asked questions and answers	Frequently asked questions and answers

* How to proceed with the recruitment (e.g., written tests, face to face interview, etc.). ^†^ The location of the department/hospital. ^§^ The number of inpatient cases or weekly outpatient cases per resident. ^||^ Independent office, independent rest room when on-call, etc.

**Table 2 healthcare-09-00773-t002:** Conditions of training program websites.

Name of Hospital	Public/Private	Criteria Matched (Percentage, %)
Northern Area (13) *		
Keelung Chang Gung Memorial Hospital and Lovers Lake Branch	Private	5 (20%)
Taipei Veterans General Hospital	Public	16 (64%)
National Taiwan University Hospital	Public	10 (40%)
Tri-service General Hospital	Public	18 (72%)
Mackay Memorial Hospital	Private	13 (52%)
Taipei Municipal Wangfang Hospital ^†^	Public	8 (32%)
Taipei Medical University Hospital	Private	8 (32%)
Shin Kong Wu Ho Su Memorial Hospital	Private	12 (48%)
Cathay General Hospital	Private	4 (16%)
Taipei City Hospital, Renai Branch and Heping Fuyou Branch	Public	4 (16%)
Taipei and Linkou Chang Gung Memorial Hospital	Private	12 (48%)
Far Eastern Memorial Hospital	Private	13 (52%)
Shuang Ho Hospital, Ministry of Health and Welfare ^†^	Public	5 (20%)
Central Area (4) *		
Taichung Veterans General Hospital	Public	12 (48%)
China Medical University Hospital	Private	5 (20%)
Chung Shan Medical University Hospital	Private	6 (24%)
Changhua Christian Hospital	Private	5 (20%)
Southern Area (5) *		
National Cheng Kung University Hospital	Public	16 (64%)
Chi Mei Medical Center	Private	16 (64%)
Kaohsiung Medical University Chung-Ho Memorial Hospital	Private	7 (28%)
Kaohsiung Veterans General Hospital	Public	10 (40%)
Kaohsiung Chang Gung Memorial Hospital	Private	8 (32%)
Eastern Area (1) *		
Hualien Tzu Chi Hospital	Private	6 (24%)

* The hospitals were divided into four areas (i.e., northern, central, southern and eastern), according to location, Council for Economic Planning and Development, Executive Yuan. ^†^ The Taipei Municipal Wangfang Hospital; Shuang Ho Hospital, Ministry of Health and Welfare were regarded as public hospitals even if these were managed by the Taipei Medical University.

**Table 3 healthcare-09-00773-t003:** Percentage of dermatology residency websites that contain publicly accessible information on the 25 list items.

Criteria	Total (*n* = 23) (%)	Public (*n* = 9) (%)	Private (*n* = 14) (%)	*p* Value
Recruitment information				0.31
Recruitment criteria	1 (4.3%)	0 (0)	1 (7.1%)	
Interview process	0 (0)	0 (0)	0 (0)	
Contact information	18 (78.3%)	8 (88.9%)	10 (71.4%)	
Department information				0.42
History of the department	20 (87.0%)	7 (77.8%)	13 (92.9%)	
Features of the department	23 (100%)	9 (100%)	14 (100%)	
Comprehensive faculty listing	23 (100%)	9 (100%)	14 (100%)	
Message from the chairman/program director	1 (4.3%)	0 (0)	1 (7.1%)	
List of current residents/fellows	13 (56.5%)	7 (77.8%)	6 (42.9%)	
Board pass rates	0 (0)	0 (0)	0 (0)	
Alumni information and outcomes	5 (21.7%)	3 (33.3%)	2 (14.3%)	
Description of environment and equipment	21 (91.3%)	8 (88.9%)	13 (92.9%)	
Social activities	0 (0)	0 (0)	0 (0)	
Information on the surrounding area	0 (0)	0 (0)	0 (0)	
Education and research				0.48
Program goals	13 (56.5%)	6 (66.7%)	7 (50.0%)	
Research activities and accomplishments	13 (56.5%)	6 (66.7%)	7 (50.0%)	
Educational resources available to residents	18 (78.3%)	8 (88.9%)	10 (71.4%)	
Conference schedule	13 (56.5%)	7 (77.8%)	6 (42.9%)	
Elective opportunities	9 (39.1%)	4 (44.4%)	5 (35.7%)	
Clinical work				0.27
Work hours	4 (17.4%)	3 (33.3%)	1 (7.1%)	
Rotation schedule	9 (39.1%)	4 (44.4%)	5 (35.7%)	
Expected case load	4 (17.4%)	3 (33.3%)	1 (7.1%)	
On-call expectation	4 (17.4%)	3 (33.3%)	1 (7.1%)	
Incentive				0.70
Salary	1 (4.3%)	0 (0)	1 (7.1%)	
Ancillary benefit	4 (17.4%)	2 (22.2%)	2 (14.3%)	
Frequently asked questions and answers				0.42
Frequently asked questions and answers	2 (8.7%)	2 (22.2%)	0 (0)	

## Data Availability

Data is contained within the article.

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
