# Peer review of "Analysis of the Content and Comprehensiveness of Dermatology Residency Training Websites in Taiwan"

_healthcare, 2021, doi:10.3390/healthcare9060773_

Round 1

Reviewer 1 Report

I congratulate the authors because I consider the topic covered in this study to be interesting, current and novel.

As the authors say, information obtained from the internet is a primary source of knowledge and highlighting the shortcomings of training centre websites can only help to stimulate their improvement and help future applicants.

The title is appropriate

Regarding the bibliography I am not in a position to comment, as I have never done an analysis of this information on the web.

What I think needs to change radically is the length. I believe that the information that the authors want to convey, that which the reader expects for the title and that which is of any interest, should take up less than half of what this text occupies.

The introduction is too long and gives a lot of data and opinions that do not contribute anything useful to the understanding of the study.

The data reflected in the tables are repeated in the text. If the tables are clear, those data should disappear from the text.

The discussion, in most of its length, is limited to repeating the results without generating a discussion as expected.

If all this is corrected, the reading becomes much more pleasant, useful and easier. And the text is reduced to less than half without losing any interest.

Tables and Figures:

-table 1, with domains and items, should make it clear which items correspond to each domain. All the items appear in a row that does not allow us to know to which domain each one corresponds.

-Figure 2 does not provide any information or facilitate understanding. I think it should be removed. Perhaps if it were designed from highest to lowest or lowest to highest it could be more useful.

Finally:

Regarding the use of the English language. I am not an expert to be able to give an opinion. But I think the text needs a specialised correction.

Author Response

Subject: Response to reviewers’ comments

June 16, 2021

Sonia Yang

Editor

Healthcare

I, along with my co-authors, would like to resubmit the attached manuscript entitled “Analysis of the Content and Comprehensiveness of Dermatology Residency Training Websites in Taiwan. The manuscript ID is healthcare-1255640.

The manuscript has been thoroughly revised in accordance with the reviewers’ suggestions. Please find our point-by-point responses to each of the comments below. All changes to the revised manuscript are noted using the “track changes” function of MS Word.

We thank you for considering our revised manuscript. We hope that the revised manuscript is now suitable for publication in your journal.

I look forward to your reply.

Sincerely,

Tzeng-Ji Chen, MD, PhD

Department of Family Medicine, Taipei Veterans General Hospital

No. 201, Sec. 2, Shih-Pai Road, Taipei 112, Taiwan

Tel: +886-2-2875-7458

Fax: +886-2-2873-7901

E-mail: tjchen@vghtpe.gov.tw

Reviewer #1

I congratulate the authors because I consider the topic covered in this study to be interesting, current and novel.

As the authors say, information obtained from the internet is a primary source of knowledge and highlighting the shortcomings of training centre websites can only help to stimulate their improvement and help future applicants.

The title is appropriate

Regarding the bibliography I am not in a position to comment, as I have never done an analysis of this information on the web.

Point 1:

What I think needs to change radically is the length. I believe that the information that the authors want to convey, that which the reader expects for the title and that which is of any interest, should take up less than half of what this text occupies.

Response 1: We have revised the manuscript based on the reviewer’s suggestions. The details of the changes are listed in the following responses. The length has been greatly reduced from about 3000 words to 2025 words.

Point 2:

The introduction is too long and gives a lot of data and opinions that do not contribute anything useful to the understanding of the study.

Response 2: Based on this comment, we have deleted the following sentence: “In the developed world, it increased from 51% to 86.6%.” We have now included data on Internet users worldwide and in Taiwan (Line 34). The sentence “Internet-based self-help interventions were promising for amelioration of distress and disease control [7,8]” was also deleted because it is not related to our topic (Line 45-46). These revisions were made in Section 1.1 Health information on the internet (Page 1).

Point 3:

The data reflected in the tables are repeated in the text. If the tables are clear, those data should disappear from the text.

Response 3: We thank the reviewer for the useful comments.

The paragraph about the criteria was deleted because this information is presented in Table 1 (Method, 2.1. Establishing the criteria, Page 4–5, Line 102–130).

We still mention the six dermatology training program websites that include over half of the criteria, but we do not list the names of the training programs. The data are displayed in Table 2 (Line 158-161).

Websites that include >80% or 0% of the items are mentioned in the manuscript for emphasis. Other items have been deleted. The data are displayed in Table 3. (Line 168-171, 174–177).

These revisions were made to Section 3.1. Website evaluation, Results.

Point 4:

The discussion, in most of its length, is limited to repeating the results without generating a discussion as expected.

Response 4: We thank the reviewer for this comment.

We deleted the repeating results and revised the whole section 4.1. Analysis of website contents to make the paragraph more fluent. (Page 9-10, Line 198-274).

In section 4.2. Comparison with research from the USA, we also revised the sentences. (Page 10, Line 286-287)

In section 4.3. Limitations, we revised some sentences to make readers realize the problems easier. (Page 10-11, Line 290-299)

Point 5:

If all this is corrected, the reading becomes much more pleasant, useful and easier. And the text is reduced to less than half without losing any interest.

Response 5: We thank the reviewer for the suggestion. The manuscript length has been reduced.

Point 6:

Tables and Figures:

table 1, with domains and items, should make it clear which items correspond to each domain. All the items appear in a row that does not allow us to know to which domain each one corresponds.

Response 6: We thank the reviewer for these comments.

We have added horizontal lines to distinguish the items under different domains (Table 1, Page 3).

In addition, we have added horizontal lines to Table 2, Table 3, and Supplemental Table A1 to present the data more clearly (Page 6–8, Page 11–12).

Point 7:

Figure 2 does not provide any information or facilitate understanding. I think it should be removed. Perhaps if it were designed from highest to lowest or lowest to highest it could be more useful.

Response 7: We thank the reviewer for their suggestion.

We have revised Figure 2 to present the data from highest to lowest.

Point 8:

Finally:

Regarding the use of the English language. I am not an expert to be able to give an opinion. But I think the text needs a specialised correction.

Response 8: The revised manuscript has been checked by an English language editing service.

Reviewer 2 Report

Internet-based resources play a pivotal role in the decision-making of students on residency. The current study was aimed at elucidating the transparency and accuracy of dermatology residency training websites in Taiwan. Not enough is known on the subject worldwide and, it seems this is the first in Taiwan.

The article is well-written and methodological.

One minor comment. The “delivery system” of the information is also of importance. Any data on website quality? (site speed, indexed pages, email privacy, mobile-friendliness extra). If not, qualitative comments on the subject from the authors will be of interest.

Author Response

Subject: Response to reviewers’ comments

June 16, 2021

Sonia Yang

Editor

Healthcare

I, along with my co-authors, would like to resubmit the attached manuscript entitled “Analysis of the Content and Comprehensiveness of Dermatology Residency Training Websites in Taiwan. The manuscript ID is healthcare-1255640.

The manuscript has been thoroughly revised in accordance with the reviewers’ suggestions. Please find our point-by-point responses to each of the comments below. All changes to the revised manuscript are noted using the “track changes” function of MS Word.

We thank you for considering our revised manuscript. We hope that the revised manuscript is now suitable for publication in your journal.

I look forward to your reply.

Sincerely,

Tzeng-Ji Chen, MD, PhD

Department of Family Medicine, Taipei Veterans General Hospital

No. 201, Sec. 2, Shih-Pai Road, Taipei 112, Taiwan

Tel: +886-2-2875-7458

Fax: +886-2-2873-7901

E-mail: tjchen@vghtpe.gov.tw

Reviewer #2

Internet-based resources play a pivotal role in the decision-making of students on residency. The current study was aimed at elucidating the transparency and accuracy of dermatology residency training websites in Taiwan. Not enough is known on the subject worldwide and, it seems this is the first in Taiwan.

The article is well-written and methodological.

Point 1:

One minor comment. The “delivery system” of the information is also of importance. Any data on website quality? (site speed, indexed pages, email privacy, mobile-friendliness extra). If not, qualitative comments on the subject from the authors will be of interest.

Response 1:

Based on our literature search, most of the previous studies did not assess the quality of the websites. Only two studies “Self-Reported Information Needs of Anesthesia Residency Applicants and Analysis of Applicant- Related Web Sites Resources at 131 United States Training Programs” and “Evaluating Dermatology Residency Program Websites” assessed quality. In the first study, website quality was subjectively scored on a 1–10 scale by the authors. In the second study, the authors used a Search Engine Optimization (SEO) tool (WooRank.com) to evaluate website quality. However, there are many software tools that can be used for evaluating websites. We tried to analyze the websites using different tools, but the results were inconsistent. In conclusion, it is difficult to assess website quality due to subjectivity and inconsistency. We hope there will be a more objective tool for evaluating website quality in the future.
